# A Comparison of Two Types of Acoustic Emission Sensors for the Characterization of Hydrogen-Induced Cracking

**DOI:** 10.3390/s23063018

**Published:** 2023-03-10

**Authors:** Dandan Liu, Bin Wang, Han Yang, Stephen Grigg

**Affiliations:** 1TWI Ltd., Granta Park, Great Abington, Cambridge CB21 6AL, UK; 2Department of Mechanical and Aerospace Engineering, Brunel University London, Uxbridge UB8 3PH, UK

**Keywords:** hydrogen-induced cracking, sensor type, Nano30, VS150-RIC, signal characteristics, source location

## Abstract

Acoustic emission (AE) technology is a non-destructive testing (NDT) technique that is able to monitor the process of hydrogen-induced cracking (HIC). AE uses piezoelectric sensors to convert the elastic waves generated from the growth of HIC into electric signals. Most piezoelectric sensors have resonance and thus are effective for a certain frequency range, and they will fundamentally affect the monitoring results. In this study, two commonly used AE sensors (Nano30 and VS150-RIC) were used for monitoring HIC processes using the electrochemical hydrogen-charging method under laboratory conditions. Obtained signals were analyzed and compared on three aspects, i.e., in signal acquisition, signal discrimination, and source location to demonstrate the influences of the two types of AE sensors. A basic reference for the selection of sensors for HIC monitoring is provided according to different test purposes and monitoring environments. Results show that signal characteristics from different mechanisms can be identified more clearly by Nano30, which is conducive to signal classification. VS150-RIC can identify HIC signals better and provide source locations more accurately. It can also acquire low-energy signals better, which is more suitable for monitoring over a long distance.

## 1. Introduction

Hydrogen-induced cracking (HIC) is a common phenomenon in the energy sector, such as in oil and gas pipelines, offshore oilfield equipment, and pressure vessels. The generation of a large amount of HIC can lead to the rupture of structures, leading to contents leakage or even an explosion [1,2,3]. Therefore, it is necessary to monitor HIC in servicing assets. Once significant HIC activities are detected, maintenance and repairs can be carried out to prevent more serious consequences. Acoustic emission (AE) technology, a non-destructive testing (NDT) method, is regarded as a suitable method for monitoring the development of defects in safety-critical structures without interfering with normal operations [4,5,6]. 

Elastic waves generated from local crack changes in structures such as HIC are captured by AE sensors and converted into electrical signals for further analysis. The acquired signals are influenced by not only the fracture mechanism, but also wave propagation through the structure and, significantly, the response of sensors [7]. Signal pattern recognition (i.e., identifying different fracture mechanisms) is needed in order to understand HIC occurrence or damage mechanisms. It is thus desirable to identify signals that can best characterize the source. This makes the selection of sensors vitally important. Most of the AE sensors used in practice have their own distinct acoustic resonance, thus the name resonant AE sensor, which means the measurement is significantly more effective at a particular frequency. Sensors are therefore typically selected based on the peak frequency of the wave. However, a large range of frequencies may exist, particularly in cases where multiple damage mechanisms are present, and the acquired signals are influenced heavily by the response of the sensors. In such cases, wideband or broadband AE sensors are more suitable. However, they often lack the sensitivity needed for practical use.

For energy pipelines, the generation and propagation of HIC is always accompanied by H_2_ evolution and corrosion activities. Signals of H_2_ evolution and corrosions are inevitably collected at the same time together with those from HIC. Thus, distinguishing different source mechanisms and extracting signals from HIC is only the first step of analysis and a vital process for improving the accuracy of source location. Commonly, experimental tests that obtained fractures of HIC are performed with passing H_2_S gas according to ANSI/NACE Standard TM0284-2003 [8]. Merson et al. [9] used wideband AE sensor MSEA-L2 to monitor HIC activities in 09GSF pipeline steel under experimental conditions using a 5% NaCl and 0.4% CH_3_COONa solution with H_2_S at a partial pressure of 1 bar. They distinguished three types of signals from H_2_ evolution, FeS film formation, and HIC, respectively, at different stages with different frequency spectra of signals. Shen et al. [10] investigated HIC development in 20R steel using AE sensor VS900-RIC by partially immersing samples in saturated solution. Three clusters of signals from background noises, FeS layers, and HIC were differentiated with classifications based mainly on the amplitude and energy. It was noted that the peak frequency ranges in the spectrum of signals were very close. Smanio et al. [11,12] adopted Nano30, a resonant sensor, for monitoring HIC tests on X65 and C110 steels. They monitored H_2_ evolution and corrosions with separate experiments and identified the characteristics of these two sources of signals. HIC signals were distinguished from the mixed signals using the duration of different signals as the main factor for separation. 

The consensus from the published results appears to indicate that three different types of signals from the H_2_ evolution, FeS layer formation in the corrosion process, and HIC development, respectively, can be acquired and clustered during tests. However, the signal source classification methods chosen by different researchers differ. One of the reasons is that different types of sensors respond to the same event differently. In open literature, only a single type of sensor was reported as being used in experiments. To the best knowledge of the present authors, no trials have been conducted specifically to compare performances of different types of sensors. One of the motivations of this study is to test two types of sensors under the same condition to better understand their advantages and shortcomings for HIC monitoring. In addition, in all reported tests, it took a long time to produce HIC, typically from 4 days to several weeks, which is a major drawback for laboratory tests. In the present study, the electrochemical hydrogen-charging method was chosen to verify that HIC can be generated in an accelerated approach [13].

Two types of AE sensors, VS150-RIC and Nano30, were chosen in this study for monitoring HIC development. The VS150-RIC sensor is a resonant narrowband sensor commonly used for monitoring steel equipment. It has a peak resonance at 150 kHz, and its frequency response ranges from 100 to 450 kHz. Therefore, the VS150-RIC sensor may respond well to HIC signals whose main frequency band is 150 to 200 kHz [9]. However, considering that the average frequency of HIC signals reported by Smanio et al. [11] is between 0 to 100 kHz, a resonant sensor with a wider response frequency was also introduced for comparison. The Nano30 sensor has a frequency range from 50 to 750 kHz with the peak frequency at 300 kHz. It also exhibits a relatively flat response from 125 to 750 kHz. The size of Nano30 is also small with a diameter of 8 mm, making it suitable for the measurement of small spots or complex geometries. The response curves from the sensors’ calibration certificates of the two types of sensors used in this study are shown in Figure 1a for Nano30 and Figure 1b for VS150-RIC. The frequency response of VS150-RIC is representative of the sensor in a face-to-face measurement setup which is based on the ASTM standard E976. The excitation was accomplished by an Olympus V103-RM transducer driven by a KeySight function generator with an output voltage of 0.1 Vrms. 

Overall, the influences of the two selected AE sensors on HIC activities are presented in this study with three aspects considered in signal acquisition, the signal source recognition, and the source location to provide a basic reference for sensor selection for monitoring HIC according to different testing purposes. In addition, the signal pattern from the different sensors can also be better understood for data processing and analyses.

## 2. Materials and Methods

Experiments were carried out on a thin carbon steel plate. Under laboratory condition, the electrochemical hydrogen-charging method was used as an effective way to obtain HIC development safely and efficiently. To better compare the signals acquired by the two types of sensors in wave propagation, source location, and other aspects, two array arrangements of sensors were used. Four sensors of the same type were first placed along a straight line to analyze the wave propagation and characteristics of the HIC signals. The four sensors were then arranged into a parallelogram array to identify and compare the source locations. Both types of sensors were tested separately in these two arrangements.

### 2.1. Tested Specimens

HIC susceptibility is considered high for a carbon steel in an environment with high sulfur content (>0.002%) in the as-rolled condition [16], with MnS inclusions believed to be the main sites of HIC. Moreover, it has been proven that HIC can be generated relatively easily in metals with a banded ferrite–pearlite microstructure [17]. Based on these, a rolled ASTM A516 Grade 65 carbon steel plate was selected. The chemical compositions of the steel are displayed in Table 1. Dimensions of the sample plates were 500 mm × 300 mm × 5 mm, with surfaces polished to 2500 grit and cleaned with alcohol.

### 2.2. Preparation for Electrochemical Hydrogen Charging

A Gill AC potentiostat (from ACM Instruments Ltd., Grange-over-Sands, UK) was used to provide a constant electric current. The specimen and platinum were set as the cathode and the anode, respectively. Electrolyte 0.5 mol/L H_2_SO_4_ and 0.5 g/L NaAsO_2_ dissolved in deionized water was used [18]. The applied current was adjusted by the Sequencer software on a computer. The reaction equation at the cathode can be described as:(1)$$2H^{+}+2e\to \left[H\right]+\left[H\right]\to 2H/H_{2}↑$$

Some of the generated hydrogen atoms recombine, form hydrogen gas, and escape. The remaining ones enter into the steel sample and migrate into hydrogen traps, which attract hydrogen atoms by surrounding strain fields, such as vacancies, dislocations, micro-voids, etc. The acidic solution was used to provide more H_2_ while NaAsO_2_ inhibits hydrogen atom recombination to promote more of them into the steel sample. The electrolyte was held in a glass, conical, funnel-like container with an outer diameter of 38 mm, which was sealed to the specimen by silicone.

Since signals from H_2_ evolution would also be acquired by the sensors, the surface area of the sample exposed to the electrolyte was reduced to a 10 mm × 10 mm square by covering the rest of the surface area with Belzona 4311 coating to reduce non-targeted signals. The current density was set at 5 mA/cm^2^ to control hydrogen bubble generation.

### 2.3. Experiments for Signal Discrimination

When the electrochemical H-charging method is used, acoustic signals are emitted from both H_2_ evolution and HIC. The signals from H_2_ evolution needs to be characterized so they can be filtered out for HIC signals. A brief electrochemical H-charging test (of 2 h) was carried out with the electrolyte containing 0.5 mol/L H_2_SO_4_. However, in this study, the activity of crevice corrosion in the boundary silicon seals also generated acoustic waves. In addition, cathodic corrosions (uniform or pitting corrosions) also occurred in the steel sample when sulfuric acid was used as the electrolyte for cathodic hydrogen-charging [19], generating more signals. Considering these, another brief test (also 2 h) without the electric current was also performed on the surface area, which generated signals from uniform corrosions. Details of these two tests are listed in Table 2. Because of the wider band of the frequency range of the generated signals, Nano30 was chosen for these two tests. Four Nano30 sensors were connected to a Vallen-GmbH AMSY-6 data acquisition system through the external Vallen AEP4 pre-amplifiers with 34 dB gain. The Loctite SI 595 silicon sealant was used as a couplant to fix the sensors on the specimen’s surface. The amplitude threshold was set at 40 dB to filter out noises. The sampling rate was 2.5 MHz and the rearm time, defined as when a sensor is ready to acquire a new hit data set, was 200 µs. From the results of these two preliminary tests, signals from H_2_ bubbles and corrosions were distinguished, allowing HIC signals to be extracted from mixed signals in later tests.

To better compare the responsiveness of the two types of sensors used in this study, particularly for the sensitivity at different distances from the same signal source, Test 3 was carried out. The setup and sensor locations are shown in Figure 2. The mixed electrolyte solution was 0.5 mol/L H_2_SO_4_ and 0.5 g/L NaAsO_2_, and the current was 5 mA/cm^2^. The test duration was 20 h. Four sensors of each type were placed on the sample surface, as shown in Figure 2b. (Note that Nano30 sensors were obscured by other devices because of their small size in Figure 2a.) Smaller circles numbered 1 to 4 represent the Nano30 sensors, while bigger ones of 5 to 8 represent VS150-RIC. The signals obtained by the VS150-RIC sensor were amplified by the internal 34 dB pre-amplifier. Magnetic holders were applied to mount the VS150-RIC and the couplant between the sensors, and the specimen’s surface was ultrasound gel. The center-to-center distance between neighboring sensors was 100 mm. Sensor 1 was placed next to the border of the corrosion area. Sensor 5 was placed on the sample plate’s back surface of the working area.

### 2.4. Experiments on Signal Source Location

To compare the source locations identified by Nano30 and VS150-RIC, Test 4 was carried out with a parallelogram sensor array, as shown in Figure 3a, the physical setup, and Figure 3b, the locations of sensors. Smaller circles numbered 1 to 4 represent the Nano30 sensors and bigger ones of 5 to 8 represent VS150-RIC. The coordinates of the sensors on the sample surface are listed in Table 3. Other parameters were the same as previous tests.

## 3. Results and Discussion about Signals Discrimination

### 3.1. Tests 1 and 2

Comparing the results of these two tests, fewer than 100 events were recorded in Test 1, whereas more than 6000 events were in Test 2, which indicates that uniform corrosion was greatly suppressed in Test 1. Nevertheless, three types of signals were present during both tests. Typical waveforms and frequency spectrums of these three types are shown in Figure 4a, Figure 4b, and Figure 4c, respectively. The quantities of each type are shown in Table 4.

Signals belonging to Type 1 have low amplitudes, short durations, with a peak frequency at about 150 kHz, and contain few contents at lower frequencies. Type 2 signals differ from Type 1 as they contain significantly more low-frequency contents between 50 to 100 kHz, and their peak frequencies spread out up to 200 kHz. Type 3 signals were present in small amounts in both tests with waveforms similar to those of Type 2, but high-frequency components accounted for a large proportion with the peak frequency generally weighted at 350 kHz. As shown in Table 2, the most different phenomena between Tests 1 and 2 should be that numerous uniform corrosion happens in Test 2 while that reaction is largely suppressed in Test 1, which corresponds to the situation of Type 2 signals. Signals of Type 2 accounted for 46.34% in Test 1, but the proportion reached 71.05% in Test 2 where large numbers of signals were obtained after 20 min, indicating uniform corrosion. Moreover, others’ works [20,21] indicated the same characteristics of uniform corrosion signals as those represented by Type 2. As for the signals of Types 1 and 3, signals of Type 1 were acquired much earlier (within 200 s) than those of Type 3 (about after 2000 s) in Test 1. This suggested that Type 1 signals are possibly from H_2_ evolution because a large amount of small hydrogen gas bubbles were observed immediately after applying the current. At the same time, in light of the results of other researchers [9,10,11,12,22,23], signals from H_2_ bubbles are of low frequencies. Therefore, signals of Type 1 are likely to be generated by H_2_ evolution while those of Type 3 are from crevice corrosion.

### 3.2. Test 3

This test was conducted for 20 h. The area in contact with the acidic solution is shown in Figure 5. Three types of physical phenomena were observed, i.e., HIC, uniform corrosion, and crevice corrosion. With the addition of H_2_ gas activity, there were in total four kinds of signal sources during the tests, which is consistent with the analysis of Section 2.3.

At the beginning of Test 3, only Sensor 5 received a large number of signals. In addition, the number of signals received by VS150-RIC was about five times those received by Nano30 at the same distance. Table 5 gives the hit number of each channel. The difference is due to the response ability of the sensors. The peak sensitivity of VS150-RIC is −28 dB (ref 1 V/uBar) while that of Nano30 is only −72 dB (ref 1 V/1 uBar). On the basis of the characteristics of each signal cluster that are summarized in Section 3.1, the three kinds of signals from H_2_ evolution, uniform corrosion, and crevice corrosion were easily classified. Figure 6, Figure 7 and Figure 8 demonstrate the typical waveforms and spectrums from the same event of these three clusters acquired by Nano30 (Figure 6a, Figure 7a, and Figure 8a) and VS150-RIC (Figure 6b, Figure 7b and Figure 8b), respectively. 

As Figure 6a shows, low energy released by H_2_ activity is not sufficient for signal acquisition by Nano30 at a distance beyond 200 mm. Some elastic waves emitted by the events of H_2_ evolution can only be acquired by VS150-RIC. It can be seen in Figure 6a,b that Sensor 6 was triggered 279.8 µs earlier than Sensor 2, while the rise time of the signals from Sensor 6 was 278.1 µs. Figure 6c shows comparison of the waveforms at the peak amplitude of the two signals which are circled by the red rectangles in Figure 6a,b. The changes of these two waveforms are the same and almost overlap, except the signal peak amplitude of Sensor 6 is higher at 0.13 mV (42.1 dB) than that of Sensor 2 at 0.11 mV (40.2 dB). This indicates that fluctuations arrived at both sensors almost simultaneously. It also shows once again that the intrinsic frequency characteristic of the signals from H_2_ evolution in this test is of a narrow-band frequency with a peak value around 150 kHz, which is independent of the sensor type. However, due to the higher sensitivity of VS150-RIC at frequency 150 kHz, Sensor 6 was triggered first to acquire this signal at a higher amplitude. In addition, the elastic wave arrival time given by Sensor 6 is more accurate, which is important for improving the accuracy of source location.

Figure 7 displays the waveforms and spectrums acquired by the Nano30 and VS150-RIC sensors of a typical event of uniform corrosion. It can be seen that the signals by Nano30 (Figure 7a) include a larger proportion of low-frequency components, with the frequency peaks of some signals in the range of 0 to 100 kHz. For the signals obtained by VS150-RIC (Figure 7b), the frequencies are largely in the range of 130 to 160 kHz, with only a smaller proportion of low-frequency components. Nevertheless, at the same distance, the amplitude of the Nano30 signals is only less than 1 dB lower than that of VS150-RIC.

The waveforms and spectrums of a typical event of crevice corrosion are shown in Figure 8. For such signals presented by Nano30 (Figure 8a), the frequency of the main component is around 150 kHz or 350 kHz with a wider range from 100 to 400 kHz. For the signals acquired by VS150-RIC (Figure 8b), the frequency distribution is still dominated around 150 kHz, accompanying with high-frequency components. As for signals from uniform corrosion, the amplitudes of the signals received by Nano30 and VS150-RIC are within a few dB. 

Except for the three kinds of signals discussed above, a new cluster of signals appeared during this test, which was likely to be from HIC, and is shown in Figure 9. It can be seen that the frequencies are all distributed in the range of 120 to 180 kHz with the peak value around 150 kHz. The spectrums are similar to those from H_2_ activities, but with a wider frequency range and higher energy. Moreover, the amplitude of the HIC signals received by Nano30 (Figure 9a) is about 0.29 mV (8 dB) lower than that of VS150-RIC (Figure 9b) at the same distance.

Some basic characteristics can be extracted from a signal, such as peak amplitude, rise time, duration, energy, counts, etc. from the time domain as well as peak frequency and spectrum gravity from the frequency domain. Aiming to monitor different situations, many scholars identified the damage mechanisms by individual or multivariate features extracted from AE signals [24,25,26,27]. In this study, signals obtained by Sensors 2 and 6 were least affected by attenuation and reflection due to the two sensors being close to the HIC area and far away from the boundary of the specimen. Therefore, the basic characteristics of signals obtained from Sensors 2 and 6 were considered to be applied for mechanism identification. As said in the discussion in Section 3.1, it can be seen that peak frequency is a key parameter for discrimination. Nevertheless, some signals from different sources express the same peak frequencies at around 150 kHz. In this case, spectrum gravity plays an important role in this study, whose value will bias towards lower or higher frequencies. As for the basic characteristics from the time domain, their values depend more or less on threshold, except for amplitude and energy. Moreover, some of them are not completely independent; for example, the values of duration and count are positively correlated, whose Pearson correlation coefficient is 0.86 in this study. As a result, redundant data will not only increase the complexity of classification, but also affect the recognition results. The coefficient of variance (CV) is a statistic that measures the dispersion of data distribution, which is always used for the comparison of variation degrees of multiple parameters [25]. A larger CV value means that the signal represented by the characteristic is more different, which is possibly more easily used for signal discrimination. The CV of AE amplitude, rise time, duration, counts, and energy in this study are displayed in Table 6. It can be seen that amplitude has little effect on signal discrimination. Although the CV of rise time is relatively higher, the rise-time values of different sources’ signals are widely and similarly distributed, which demonstrates that it is not suitable for signal discrimination. It is enough to choose one between duration and counts because of the strong correlation. Therefore, duration, energy, peak frequency, and spectrum gravity were chosen as the main characteristics for signal discrimination, which is also consistent with others’ work [9,12].

Values of the main characteristics of all AE signals acquired by Sensors 2 and 6 are summarized in Table 7. Comparing the signals obtained by the two types of sensors, the frequency distribution of the signals acquired by Nano30 is variable. For different signal sources, there are three different frequency peaks. Regardless of which event the signal is emitted from, the spectrum displayed by VS150-RIC is all dominated at 150 kHz, accompanied by fewer components in other frequency ranges. For signals at 150 kHz, the waveforms and spectrums from the two types of sensors are identical. In spite of that, VS150-RIC is shown to obtain a more complete signal waveform with higher amplitudes. Even at a long distance, VS150-RIC still acquired signals from low-energy events while Nano30 was not triggered. 

However, for signals that contain components in other frequency ranges, the waveforms acquired by the two types of sensors are clearly different whilst still having similar amplitudes. VS150-RIC does lose significant amounts of low frequencies in the range below 100 kHz, so signals acquired by Nano30 are more accurate in the low frequency range. From Table 7, the signals from uniform and crevice corrosions can be identified just from the peak frequency and the spectrum gravity. HIC signals can then be differentiated with the value of the signal duration or energy. However, some of the extracted featured values by VS150-RIC for different events were close, which is not conducive to signal classification to distinguish HIC signals from other sources. For example, it is difficult to distinguish uniform corrosion signals from HIC signals based on the values of the peak frequency and spectrum gravity. Besides, the values of the duration or energy of HIC signals are relatively higher but are completely covered by those values of uniform corrosion signals.

## 4. Source Location

Discussions above reveal that signals acquired by Nano30 show different characteristics for those from different sources and thus are better for classification. However, the acquired signals may not be complete, especially from low-energy activities, resulting in inaccuracy in source location. There is a need to determine signal propagation velocities for source location.

### 4.1. Velocity Calculation

Based on Test 3, the differences in the arrival time and the distance between different sensors were applied for velocity calculations. Note that the arrival time displayed in the acquisition system is the trigger time when the signal amplitude reaches 40 dB, which may be different from the real arrival time of the signal when the signal-to-noise ratio (SNR) is low. Therefore, the Akaike Information Criterion (AIC) [28] was applied to determine the signal arrival time. Based on the autoregressive process (AP), the AIC picker was used to identify the onset point by finding the separation point of noise and signal through the minimized AIC value [29]. The arrival time picked by the AIC method is often more accurate than that determined by the threshold-crossing method. It is a reliable onset-time determination tool that can improve localization accuracy [30,31]. Figure 10 shows the result of the arrival time determined for one of the HIC signals using the AIC method. It shows −58.8 µs as the signal arrival time by AIC compared to 0 µs by the threshold-crossing method. 

Velocity results from the two types of sensors are represented in Figure 11a (Nano30) and Figure 11b (VS150-RIC) after applying AIC. It can be seen that the wave propagation velocity is distributed in the range of 100–5000 mm/ms and concentrated around 3000 mm/ms, especially for high-amplitude signals. Inaccuracy in the arrival time of some signals still exists because of incomplete waveforms, noises, wave reflection, etc. Velocities calculated with high-amplitude signals are more accurate because of more complete signal waveforms. 

In thin plates, elastic waves are of the Lamb type which exhibits a multimodal nature and velocity dispersion, with the velocity depending on the frequency and material [32]. Lamb waves can be separated into the symmetrical type (S type) and the antisymmetric type (A type) according to the particle vibration directions. Both types can be further divided into several modes due to different phase velocities, such as S0 and S1, A0 and A1, etc. to represent different orders (i.e., modalities) [33,34]. Figure 12 illustrates the dispersion curve of wave propagation with S0, A0, and A1 modes in a carbon steel plate of 5 mm thickness. According to the dispersion curve, the signals were triggered by mode A0 and the propagation velocity at 150 kHz is about 3110 mm/ms.

### 4.2. Source Location

Signals from the four different mechanisms (H_2_ evolution, uniform corrosion, crevice corrosion, and HIC activity) were divided based on the characteristics summarized in Section 3.2. Note that as shown in Test 4, each VS150-RIC acquired around 700 signals while only about 300 were acquired by every Nano30. Only events with signals captured by all four sensors of each type were retained for source location analysis. Based on this filtering approach, there were 94 and 55 HIC events obtained by VS150-RIC and Nano30, respectively. 

After applying the AIC method to all signals for the arrival time, these 149 HIC events in total were identified for their source located by the Simplex algorithm [35,36], which is an iterative approach to solve nonlinear equations. In this study, x = 150, y = 250 was used as the initial basic feasible solution. The optimal solution was then found by calculating the location errors of the vertices (which are the locations of four sensors) by iterative operations. 

The obtained source location results are shown in Figure 13. Red dots represent the source location results from the Nano30 array and blue dots represent the VS150-RIC array. The green square (145 < x < 155, 245 < y < 255) shows the hydrogen charging area in the experiments. In addition, theoretically, all calculated signal source locations should be inside the square.

For parallelogram sensor arrays, besides the inaccuracies caused by incomplete waveforms, noises, wave reflection, etc., the different propagation directions of the signals also affect the source location accuracy. As one of the purposes of this study is to macroscopically compare the localization results of the two types of sensors, events located inside the hydrogen charging square are considered correct. 

The correct rate of source location by the VS150-RIC sensor array is found to be 54.26%, while that of Nano30 is only 27.27%. This is likely due to the higher sensitivity of VS150-RIC, leading to a better signal-to-noise ratio, especially for signals at 150 kHz. Due to this feature, VS150-RIC sensors are able to respond to more signals from HIC activities than Nano30 sensors are, even from small cracks with low energy. Signals expressed by VS150-RIC are more complete in the case of rearm time being 200 µs. The determined onset time is closer to the real arrival time, which is conducive to improving the accuracy of source location.

## 5. Conclusions

In this study, two types of sensors, Nano30 and VS150-RIC, were used to monitor electrical hydrogen-charging tests. In view of the occurrence of HIC, the obtained signals were analyzed and discussed in the aspects of signal acquisition, signal clustering, and source location. It is found that in this test setup:Signals from HIC events obtained by the Nano30 and VS150-RIC sensors are similar. The typical characteristics of this kind of signals from both sensors are long duration and high energy;Frequency distributions are in the range of 120 to 180 kHz with the peak frequency at around 150 kHz;VS150-RIC sensors can acquire more low-energy signals due to higher sensitivity, especially to signals from H_2_ evolution and small hydrogen-induced cracks with a peak frequency around 150 kHz;VS150-RIC is more suitable for long-distance monitoring and signal source locating due to more accurate prediction of the wave arrival time;Signals expressed by Nano30 are closer to the real ones, providing a wider band frequency range than those by VS150-RIC;Significantly distinct characteristics can be extracted from signals obtained by Nano30 sensors for different kinds of events, enabling better signal classification, especially for automatic classification.

In conclusion, this study provides certain basic information of signals from different sources by the two types of sensors during HIC monitoring. However, the outcomes are based on the test settings arranged in the study. 

The materials and the specimen geometry heavily influence signal propagation and thus, the results. These are potentially for future research to explore. The present study provides a useful reference to signal classification and sensor selection for HIC monitoring.

## Figures and Tables

**Figure 1 sensors-23-03018-f001:**
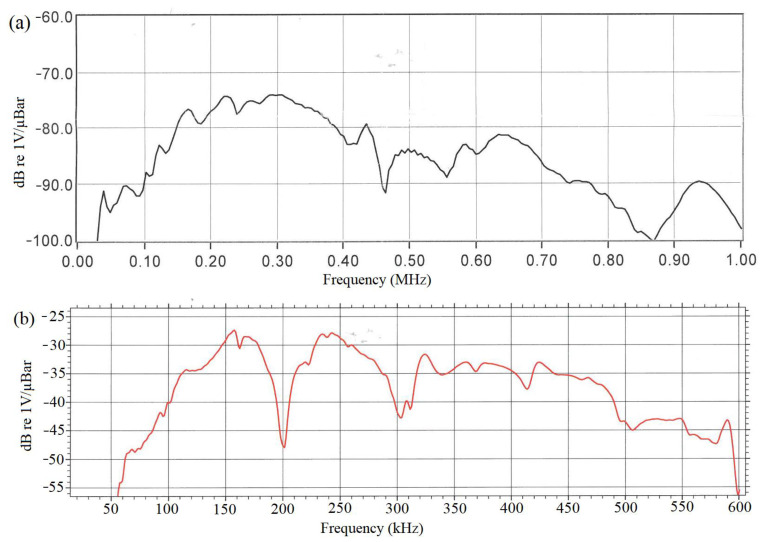
Response curves of two types of sensors used in this study: (**a**) Nano30 [14] and (**b**) VS150-RIC sensors [15].

**Figure 2 sensors-23-03018-f002:**
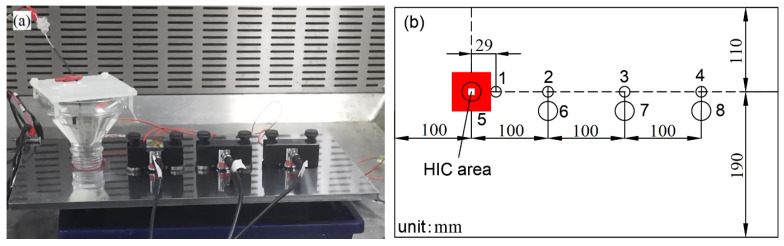
Experimental setup: (**a**) the location of sensors; (**b**) sensor positions. Nano30 locations are shown at 1 to 4, VSI50-RIC at 5 to 8.

**Figure 3 sensors-23-03018-f003:**
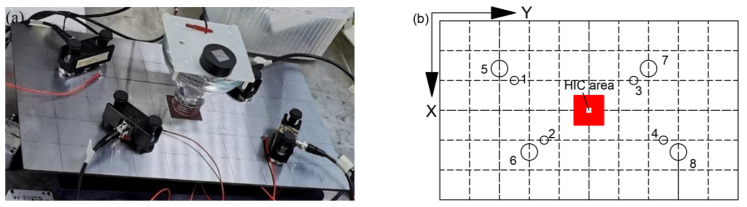
Test 4 (**a**) physical setup; (**b**) locations of sensors. Small circles represent Nano30 sensors and big circles represent VS150-RIC sensors.

**Figure 4 sensors-23-03018-f004:**
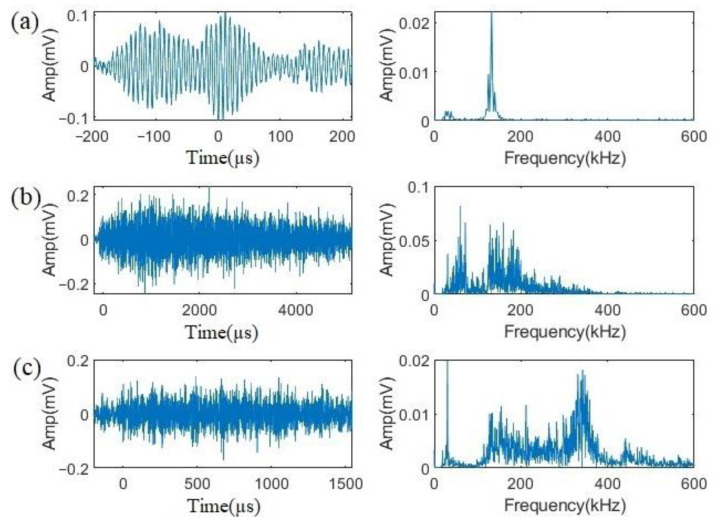
Typical waveform and frequency spectrum of signals: (**a**) Type 1, (**b**) Type 2, (**c**) Type 3.

**Figure 5 sensors-23-03018-f005:**
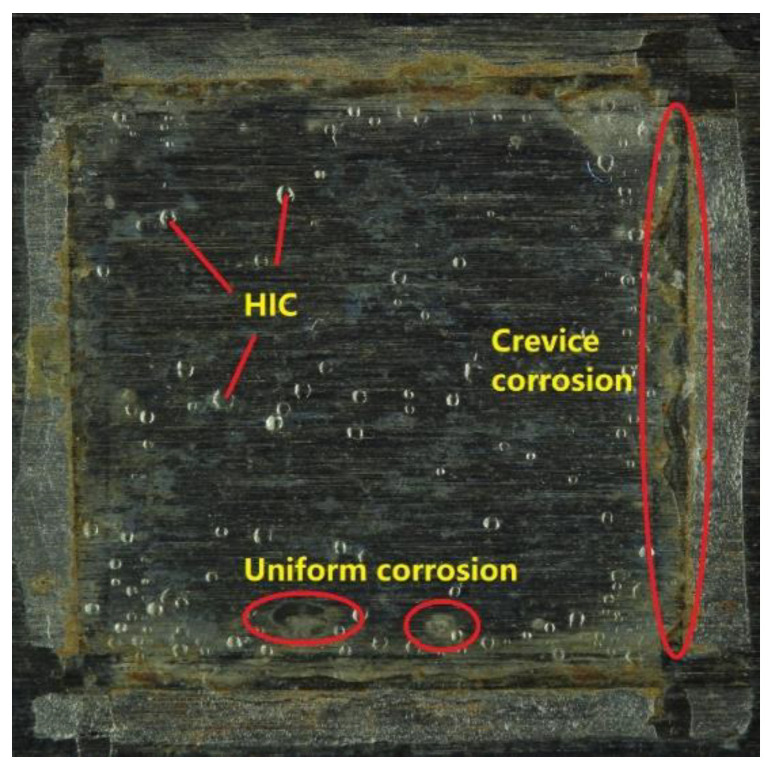
Observation of HIC area surface after tests.

**Figure 6 sensors-23-03018-f006:**
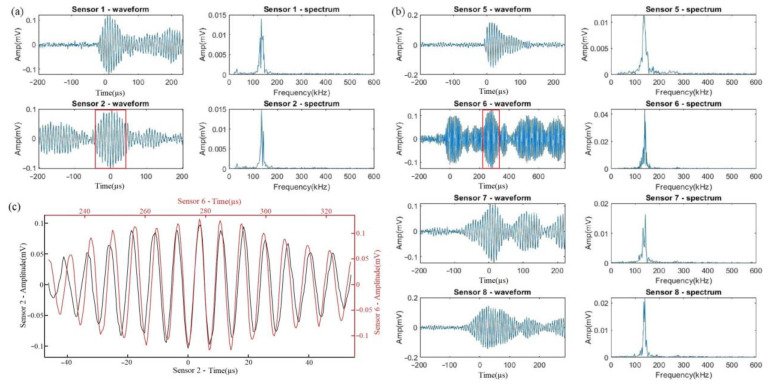
The waveforms and spectrums from the same event of H2 evolution acquired by (**a**) Nano30 sensors; (**b**) VS150-RIC sensors. (**c**) Partial waveform comparison at the amplitude peaks of signals from Sensors 2 and 6.

**Figure 7 sensors-23-03018-f007:**
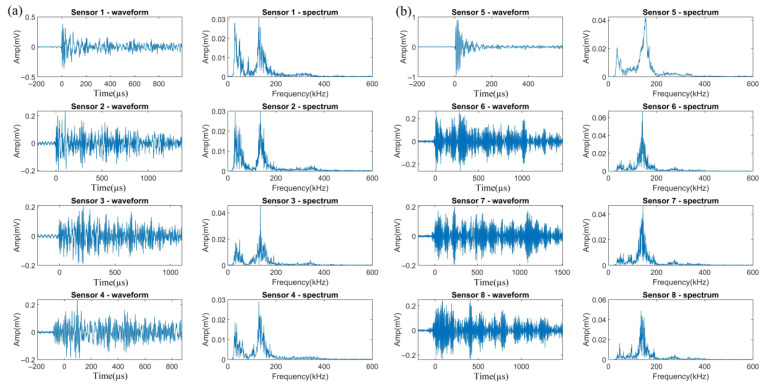
The waveforms and spectrums from the same event of uniform corrosion acquired by (**a**) Nano30 sensors and (**b**) VS150-RIC sensors.

**Figure 8 sensors-23-03018-f008:**
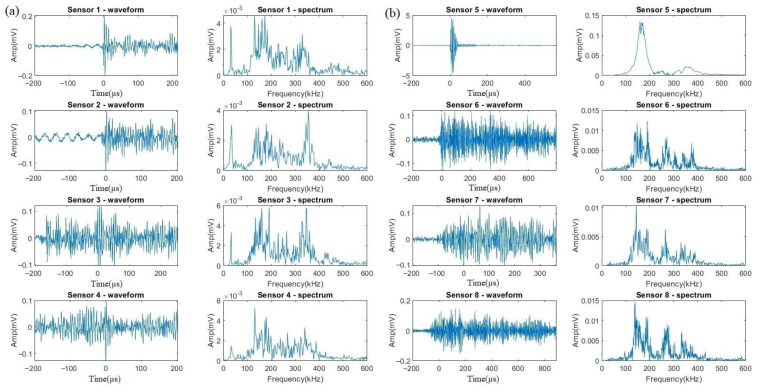
The waveforms and spectrums from the same event of crevice corrosion acquired by (**a**) Nano30 sensors and (**b**) VS150-RIC sensors.

**Figure 9 sensors-23-03018-f009:**
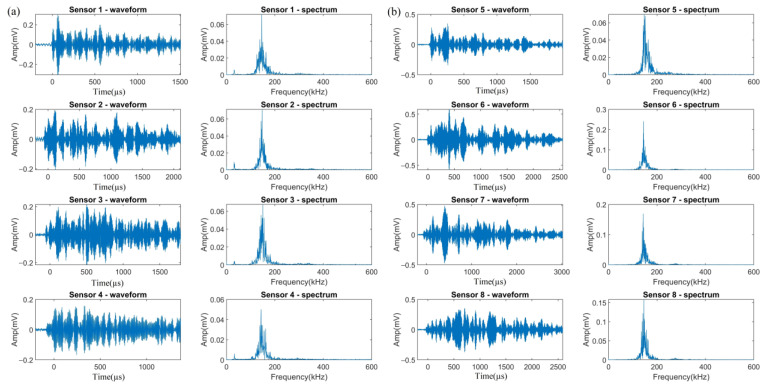
The waveforms and spectrums from the same event of HIC acquired by (**a**) Nano30 sensors and (**b**) VS150-RIC sensors.

**Figure 10 sensors-23-03018-f010:**
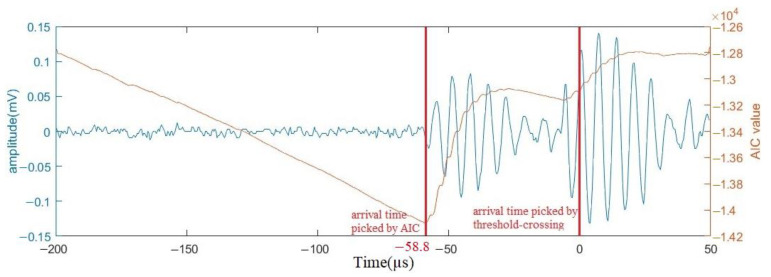
Determination of arrival time by the AIC method; threshold-crossing prediction is indicated as time = 0.

**Figure 11 sensors-23-03018-f011:**
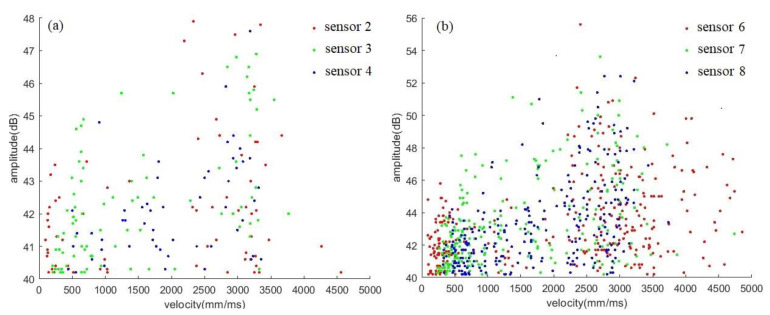
Result representation of velocity–amplitude after applying AIC: (**a**) Nano30 sensors; (**b**) VS150-RIC sensors.

**Figure 12 sensors-23-03018-f012:**
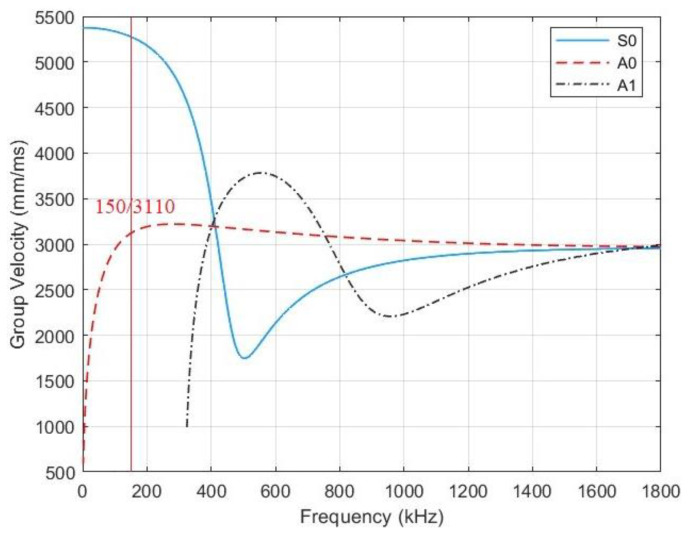
Dispersion curve on a carbon plate with the thickness of 5 mm.

**Figure 13 sensors-23-03018-f013:**
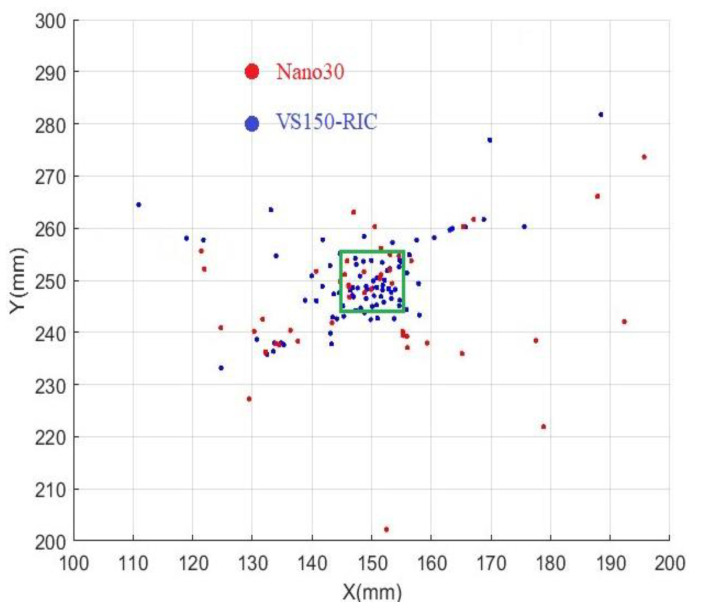
Results of source location of HIC events. Red dots: Nano30, blue dots: VS150-RIC. The green box is the charging area.

**Table 1 sensors-23-03018-t001:** Chemical compositions (%) of A516 carbon steel.

**Composition**	**C**	**Mn**	**Si**	**P**	**S**	**Cr**
Content (%)	0.137	1.073	0.169	0.011	0.005	0.015
**Composition**	**Ni**	**Mo**	**Cu**	**Al**	**Ti**	**Nb**
Content (%)	0.008	0.002	0.016	0.033	0.001	0.014

**Table 2 sensors-23-03018-t002:** Details of two short-time tests.

	Electrolyte	Current	Theoretically Signal Sources
Test 1	0.5 mol/L H_2_SO_4_	5 mA/cm^2^	H_2_ bubble + crevice corrosion + uniform corrosion (possible)
Test 2	0.5 mol/L H_2_SO_4_ + 0.5 g/L NaAsO_2_	×	H_2_ bubble + crevice corrosion + uniform corrosion (numerous)

**Table 3 sensors-23-03018-t003:** Details of sensors for Test 4.

Number	Type	Coordinates (mm)	Number	Type	Coordinates (mm)
1	Nano30	(100, 125)	5	VS150-RIC	(80, 100)
2	Nano30	(200, 175)	6	VS150-RIC	(220, 150)
3	Nano30	(100, 325)	7	VS150-RIC	(80, 350)
4	Nano30	(200, 375)	8	VS150-RIC	(220, 400)

**Table 4 sensors-23-03018-t004:** Events quantity of each signal type.

Quantity of Events	Type 1	Type 2	Type 3
Test 1	25	38	19
Test 2	1641	4756	297

**Table 5 sensors-23-03018-t005:** The hit number of each channel.

Sensor	1	2	3	4	5	6	7	8
Hit number	306	75	95	56	34,824	391	290	326

**Table 6 sensors-23-03018-t006:** Coefficients of variance (CV) of AE amplitude, energy, duration, counts, and energy.

	Parameters	Amplitude	Rise Time	Duration	Counts	Energy
CV						
Signals from Sensor 2		4.78%	104.62%	154.75%	154.43%	108.63%
Signals from Sensor 6		6.16%	130.45%	143.53%	121.16%	152.72%

**Table 7 sensors-23-03018-t007:** Main characteristics of AE signals related to different sources.

	Parameters	Duration (µs)		Energy (eu)		Peak Frequency (kHz)		Spectrum Gravity (kHz)	
Sources		Nano30	VS150	Nano30	VS150	Nano30	VS150	Nano30	VS150
H_2_ evolution		<600	<1000	<100	<300	130–150	130–150	130–150	130–150
Uniform corrosion		Distributed in the whole range				30–150	130–160	~100	120–150
Crevice corrosion		<1000	<1500	<200	<1000	>180	150–300	200–300	180–300
HIC		>600	>1000	>700	>900	120–180	120–180	120–180	120–180

## Data Availability

The data presented in this study are available on request from the corresponding author.

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
