# Peer review of "A Comparison of Two Types of Acoustic Emission Sensors for the Characterization of Hydrogen-Induced Cracking"

_sensors, 2023, doi:10.3390/s23063018_

Round 1

Reviewer 1 Report

The proper selection of acoustic emission (AE) sensor is essential for online monitoring and accurate characterization of mechanical damage. This work investigated the monitoring of HIC processes using two commonly used AE sensors (Nano30 and VS150-RIC) and compared the AE signals obtained from the two sensors. The objective of this work is important and the results are properly discussed. I believe the readers in Sensors will be very interested in the topic of this paper. However, the major drawback of this paper is lack of scientific novelty. Below are several specific comments that might be helpful for improving the overall quality of the paper.

1.      It is necessary to provide the frequency response curves of the two sensors because this paper aims to compare the AE signals acquired from the two different sensors.

2.      In Fig. 3(a-c), are you sure the unit of time of each waveform is second? Why the length of each waveform is not the same?

3.      It is not clear how the three types of AE signals are discriminated. Did you use some clustering methods such as K-means?

4.      In line 262, the unit of amplitude of AE signals is expressed in “dB”. However, the unit in Fig.8 is “mV”. Please keep the consistency.

5.      There are many published papers demonstrating the frequency results of AE signals generated from electrochemical corrosion processes such as pitting, uniform corrosion, etc. The authors can refer to the published papers and compared their results with yours.

https://doi.org/10.3390/cmd2010001

https://doi.org/10.2320/matertrans.M2014373

6.      There are numerous AE parameters with different physical meanings that can characterize the development of microstructural damage and therefore the multi-parameter analysis is very important for understanding the development of AE sources. In Table 6, the authors provide a statistic results of a few parameters of AE signals corresponding to different sources, which is not enough. Some important parameters such as count, peak amplitude, RA value, information entropy are neglected. I suggest refer to the following papers and explore more AE parameters and then discuss which parameters are more suitable to distinguish the AE signals related to different sources.

https://doi.org/10.1016/j.ijfatigue.2022.106860

http://dx.doi.org/10.1016/B978-1-78242-381-2.00010-9

Reviewer 2 Report

A very interesting article showing the differences in the use of two different types of AE sensors. The article is written in simple and understandable language. The conclusions are correct and reflect the essence of the study. The given values and ranges of changes for individual parameters characterizing the recorded AE signals. The authors could indicate all the parameters and only then choose those of the greatest diagnostic importance. Of course, when considering signals from various phenomena such as crevice corrosion, uniform corrosion or HIC, these parameters may differ, which the authors rightly noticed.

The methodology lacks information on the method of mounting the sensors to the tested sample. Only magnetic holders for VS150-RIC sensors are shown in Fig. 1. This information should be in the text, as well as a fixing method for Nano-30 sensors. There is no information whether and, if so, what kind of coupling agent was between the sensors and the sample surface. There is no information whether the same preamplifiers were used for both types of sensors.

In all article symbol “us” should be changed into “µs”.

I recommend publishing the article with minor corrections.

Reviewer 3 Report

Dear author
AE technique is one of the most important techniques in monitoring materials. And the novel side of this study is using two different sensors on detecting the activities. The content is very good. However, quality of paper presentation can be improved by playing tables looks because they look so simple which can be design as two rows. There are some recommendation on uploaded file. Please check those.

The study addresses how acoustic activities are considered in case of using two different sensors. Since  the paper claims that it is the only study so far which used two different sensors to detect acoustic activity the differences, this is the original side of this study. A sensor can be more precise compared to the other one. This allows the readers to see using the right sensors based on service area. These sensors can be tried with other materials to see and compare the current results as future studies. The conclusions are consistent and addresses the main questions. The references are related with the topic and they are all about acoustic studies. The tables seems so simple, especially Table 1 which can be design as two rows, currently too long for its row. Figures are okay but Figure 5c which is slightly bigger than other which destroys the representability of the others. The AE unit should be consistent through the paper, either dB or mV.
Best regards.

Round 2

Reviewer 1 Report

I have no other comments.